# INFORMATION GAP IN CHAIN-OF-THOUGHT INDUCES IMPLICIT THINKING THAT FAILS IN LENGTH GENERALIZATION

## ABSTRACT

Recent works reveal that Chain-of-Thought may not faithfully reflect the model's actual reasoning, as the semantics can diverge from model's underlying "implicit thoughts". In this work, using a synthetic dataset with controllable complexity, we find signs of *implicit thinking* in models after supervised finetuning (SFT) on CoT rationales, that is, the models have internally identified all necessary variables to be solved before generating the actual CoT. This implicit thinking ability sharply degrades as the required CoT steps exceed those seen during training, hence preventing the model from generalizing to more complex problems. To understand why implicit thinking emerges during SFT on explicit CoT rationales, we first define "information gap" within a CoT based on the ratio of unexplored actions and all admissible actions at each state. We hypothesize that a large information gap (a lot of admissible but unexplored actions) force LLMs to justify the actions explored in golden CoT by looking for clues in its internal representation, hence leading to implicit thinking. We benchmark 4 types of CoT, each based on a different graph traversal heuristic, and observe a positive correlation between the magnitude of information gap in CoTs and the implicit thinking ability in models finetuned on these CoTs. We further support this hypothesis by showing that actively reducing the information gap by including multiple CoT trajectories per question can reduce implicit thinking and enhance generalization to more complex questions. Overall, our findings suggest rethinking the role of CoT in LLM reasoning and understanding the necessary condition of learning generalizable CoT.[1]

## 1 INTRODUCTION

Chain-of-thought (CoT) enables large language models (LLMs) to generate a sequence of intermediate reasoning steps in natural languages before predicting a final answer (Wei et al., 2022). It is the foundation of recent advancement of LLM in reasoning-heavy tasks such as solving olympiad-level math and coding problems. Researchers (Baker et al., 2025) have also argued that by monitoring the generated CoT rationales, humans, or other models can better interpret the "thinking process" of LLMs and hence reliably verify the soundness of machine reasoning.

However, findings from recent work challenge the aforementioned interpretation of CoT as the human-like "thoughts" of LLMs: Sun et al. (2025) show that models trained with SFT cannot exploratively generalize to solve more complex problems requiring the same set of knowledge as the training data; while another line of work (Arcuschin et al., 2025; Stechly et al., 2025; Barez et al., 2025) finds that CoT sometimes is a post-hoc rationalization of the *implicit* thinking already done by LLMs: it only reveals partial thinking processes or even has little to no causal effects on the final predicted answer. Building upon these previous findings, we hypothesize that LLM's ability to generalize to more complex problems is negatively correlated with the amount of implicit thinking they perform prior to generating the CoT. We further study which data factors in CoT supervision give rise to implicit thinking and whether we can control it to unlock more generalizable reasoning.

To isolate the core reasoning ability from confounders like domain knowledge/tool-usage and to eliminate the chance of data contamination in evaluation, we synthesize WORLDOFBOXES (WOB),

---

[1]We aim to open-source our code and data upon publication.

a grade-school-level math dataset following the same principles used for iGSM (Ye et al., 2024): as shown in Fig. 1, for each question, we randomly create a dependency tree graph of "boxes" and assign every box a weight between 0 to 23 randomly. The question asks for either the weight of a source box at leaf given only the weight of the target box at root (R2L) or the weight of the source box at root given only the weight of the target box at leaf (L2R). Solving a question in WOB requires searching for the path connecting the source box to the target box and performing addition/subtraction between positive integers less than 24 to calculate boxes' weight. Compared to iGSM, our WORLDOFBOXES further encompasses a significantly simplified parameter structure (e.g., with only one type of instance parameter: the weight of boxes) and a considerably larger graphs (e.g., as many as 800 parameters/boxes as opposed to 28 in iGSM-hard). This allows us to evaluate generalization of reasoning within complex, unseen environments with a large number of possible states but without confounding factors from pretrained knowledge.

First, to understand LLM's reasoning ability on WORLDOFBOXES, we finetune 3 different base models of 7B parameters using different types of CoT rationale: (1) FORWARD-COT that only solve the boxes along the ground-truth path connecting the source to the target box, (2) BACKTRACK-COT that first backtracks from the target box to the source box, (3) RS-COT that traverses the entire dependency graph in random order, and (4) DFS-COT that traverses the graph following a depth-first-search procedure. We then supervised-finetune (SFT) pre-trained language models on WORLDOFBOXES questions with at most 5 layers in its dependency graph and evaluate them on test questions with as many as 8 layers. On WOB-R2L, we observe that while all models reach perfect accuracy on in-distribution (ID) test questions of 5 layers, models trained with FORWARD-COT degrades sharply on out-of-distribution (OOD) questions of more than 5 layers. Search-enabled models trained on RS-COT or DFS-COT score significantly better results on OOD questions, while BACKTRACK-COT models achieves almost perfect generalization to questions with dependency graphs up to 8 layers.

Next, to investigate how implicit thinking emerges during CoT learning and hampers generalization, we train a linear probe on frozen models' internal representations to predict whether a box is necessary[2] (e.g., box PRU in Fig. 1) in computing the target box's weight or not (e.g., box FEB). On top of FORWARD-COT models, the linear probe achieves $> 95\%$ accuracy on ID questions, revealing the fact that, before generating the first token in CoT, the model has already implicitly identified a complete list of necessary boxes. However, the probe accuracy drops dramatically on OOD questions of deeper dependency graphs. This indicates that the *implicit thinking* ability cannot length-generalize and potentially explains the FORWARD-COT models' catastrophic degradation when facing OOD questions: they rely on their implicit reasoning, not CoTs to find the path connecting the source and target box. On the other hand, RS-COT and DFS-COT models show less *implicit thinking* as the probe's accuracy is significantly lower on ID questions. On BACKTRACK-COT models, the probe's accuracy stays at random chance. These findings overall show a negative correlation between their implicit thinking ability and the generalization to more complex questions.

To understand why implicit thinking emerges during SFT on explicit CoT rationales, we propose a hypothesis explaining that language models acquire implicit reasoning ability when there exist *reasoning gaps* between CoT steps. We then show a positive correlation between the magnitude of the information gap within training-set CoTs and the implicit thinking ability measured by probing accuracy. Finally, we show that a recently proposed SFT improvements, DFT (Wu et al., 2025), in fact closes the *reasoning gaps* by scaling down the loss of off-policy CoT tokens. Empirically, we observe a significant drop in the probing performance in after applying the DFT objective. Hence, we hypothesize that a potential reason behind DFT's successes on real-world reasoning tasks is that it effectively suppress the learning of implicit thinking from *information gaps* within CoT rationales.

The contributions of this work are: (1) we introduce how we create the WORLDOFBOXES dataset and different types of CoT rationales (Sec. 3); (2) we show LLMs' failure in generalization on WORLDOFBOXES and conduct a probing analysis that exposes these models' implicit thoughts (Sec. 4); (3) we propose a hypothesis that "information gap" within CoT rationales induces implicit thinking support it with empirical evidence (Sec. 5). While we refrain from overclaiming that implicit thinking causally hinders LLMs from generalizing, we present multiple pieces of evidence that suggests a negative correlation between these two factors. Overall, our findings indicate rethinking the role of CoT (especially in SFT) in achieving generalizable reasoning.

---

[2]An unknown box is necessary if it is the ancestor of the target box.

## 2   RELATED WORK

**The faithfulness of Chain-of-Thought.**   Chain-of-Thought rationale is widely regarded as a interpretability tool (Wei Jie et al., 2024) that reveals the reasoning process of LLMs and an extension to their reasoning boundary (Zhou et al., 2023). However, more recent work (Turpin et al., 2023; Chen et al., 2025) finds that CoT sometimes is not totally faithful to the model's underlying thinking process. Sometimes it is merely a post-hoc rationalization of the *implicit* thinking already done by LLMs: it only reveals partial thinking processes or even having little to no causal effects on the final predicted answer (Arcuschin et al., 2025; Stechly et al., 2025). Most notably, Barez et al. (2025) summarize the evidence of unfaithful CoT in a wide range of work mentioned above and beyond, and challenge the soundness of treating CoT as being sufficient for interpretability.

**Probing for internal reasoning of LLMs.**   In order to "read models' mind" and expose the internal reasoning behavior that may deviate from their generated CoT rationales, previous works have leveraged probing for different indicator quantities from models' intermediate representations. Afzal et al. (2025) show that a probe can predict a model's success before it generates the first token in CoT, while another work (Dong et al., 2025) uses linear probe to successfully predict global structure attributes (e.g., response length, reasoning steps) in the future. Most relevantly, Ye et al. (2024) use v-probing to find that language models trained with SFT on synthetic CoTs already know the full list of necessary parameters before generating the CoT.

**The learnability and generalizability of CoTs in synthetic reasoning tasks.**   A number of previous works (Liu et al., 2022b; Kazemi et al., 2023; Feng et al., 2023; Wang et al., 2024; Mirzadeh et al., 2025; Shojaee et al., 2025; Malek et al., 2025) have studied the learnability of CoTs in solving reasoning-heavy tasks and the undesired shortcuts (e.g., memorization (Zhang et al., 2025b), implicit thinking (Liu et al., 2022a; Qin et al., 2025)) that arise from models' trying to imitate demonstrations of reasoning traces. Minegishi et al. (2025) extract reasoning graph by clustering hidden-state representation of CoT steps and reveal the relationship between the graph topology and underlying reasoning ability. Mirtaheri et al. (2025) compare sequential versus parallel scaling CoTs on the graph connectivity problem. Most notably, Bachmann & Nagarajan (2024) demonstrate that teacher-forcing can let the model overfit "lookahead tasks" similar to our WOB and fail in generalization. Abbe et al. (2024) show that learning knowledge extraction and simple multihop reasoning is more challenging for problems with larger "branching factors" (e.g., the number of possible next steps) compared to those with a smaller action space. Zhang et al. (2025a) further reveal that for the same problem, reasoning in the direction with lower conditional entropy is always easier for language models. A recent benchmark OMEGA (Sun et al., 2025) includes olympiad-level questions that require applying learned problem-solving skills to more complex instances. They show that LLMs cannot, for example, count rectangles in an dodecagon after trained to count in octagon, similar to our finding regarding the failure in length generalization on WORLDOFBOXES after SFT.

## 3   SYNTHETIC MATH DATASET

To isolate the core reasoning ability from confounders like domain knowledge and tool-usage in evaluation, we synthesize a grade-school math dataset called WORLDOFBOXES (WOB). It requires only commonsense (e.g., if the prompt in Fig. 1 states that box PRU weighs 1 pound less than box RYH and the box RYH weighs 22 pounds, then the models needs to generate $W_{PRU} = 22 - 1 = 21$ to solve the weight of box PRU) and addition/subtraction between positive integers less than 24.

### 3.1   CONSTRUCTING A WORLD OF BOXES FROM A DEPENDENCY GRAPH

In WORLDOFBOXES, the prompt of every data point describes a unique imaginary world made of boxes only. We first build a tree of a certain depth (as shown in the right panel of Fig. 1) as the dependency graph between all boxes with every tree node representing a box. During the creation of the tree, we randomly select the branching factor [3] of each node from the range 1-4. We then assign every box a random integer weight between 0 to 23 pounds and a unique name made by a random permutation of three capital letters (e.g., "box FEB"). The prompt (as shown in the left panel of Fig. 1) describes the relationship between every two connected boxes (e.g., "box PRU weighs 1

---

[3]A tree node with k descendants has a branching factor of k.

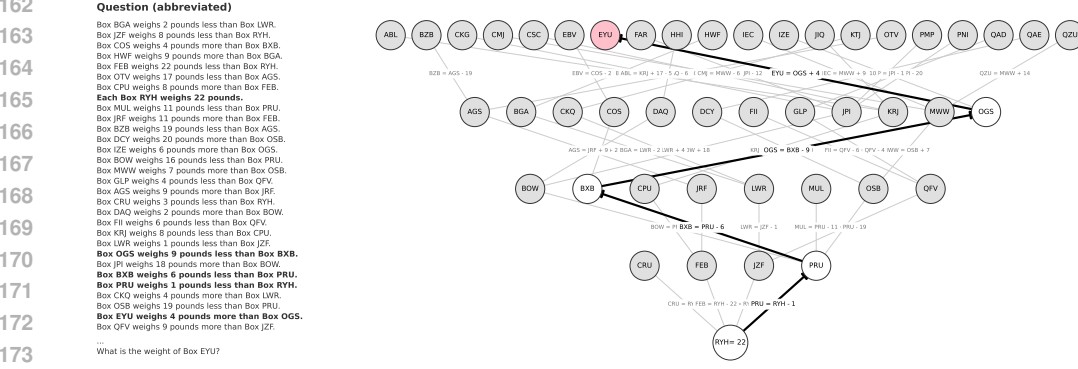

**Question (abbreviated)**

Box BGA weighs 2 pounds less than Box LWR.
Box JZF weighs 8 pounds less than Box RYH.
Box COS weighs 4 pounds more than Box BXB.
Box HWF weighs 9 pounds more than Box BGA.
Box FEB weighs 22 pounds less than Box RYH.
Box OTV weighs 17 pounds less than Box AGS.
Box CPU weighs 8 pounds more than Box FEB.
**Each Box RYH weighs 22 pounds.**
Box MUL weighs 11 pounds less than Box PRU.
Box JRF weighs 11 pounds more than Box FEB.
Box BZB weighs 19 pounds less than Box AGS.
Box DCY weighs 20 pounds more than Box OSB.
Box IZE weighs 6 pounds more than Box OGS.
Box BOW weighs 16 pounds less than Box PRU.
Box MWW weighs 7 pounds more than Box OSB.
Box GLP weighs 4 pounds less than Box QFV.
Box AGS weighs 9 pounds more than Box JRF.
Box CRU weighs 3 pounds less than Box RYH.
Box DAQ weighs 2 pounds more than Box BOW.
Box FII weighs 6 pounds less than Box QFV.
Box KRJ weighs 8 pounds less than Box CPU.
Box LWR weighs 1 pounds less than Box JZF.
**Box OGS weighs 9 pounds less than Box BXB.**
Box JPI weighs 18 pounds more than Box BOW.
**Box BXB weighs 6 pounds less than Box PRU.**
**Box PRU weighs 1 pounds less than Box RYH.**
Box CKQ weighs 4 pounds more than Box LWR.
Box OSB weighs 19 pounds less than Box PRU.
**Box EYU weighs 4 pounds more than Box OGS.**
Box QFV weighs 9 pounds more than Box JZF.
...
What is the weight of Box EYU?

Figure 1: An example in WORLDOFBOXES-R2L dataset that requires solving the weight of a target leaf box (shown in red) by finding the path connecting it and the only box (root) with known weight. Every question (shown abbreviated on the left) describes a unique world of boxes with its underlying dependency graph (shown on the right). Each statement in the question either describes the weight of the source box or the relationship between two connected boxes. Those bold statements in the question describe the path connecting the source box RYH to the target box EYU. In the dependency graph, the boxes that are NOT on this path are shaded in grey.

pound less than box RYH") and reveals the exact weight of one source box (e.g., "box RYH weighs 22 pounds"). Based on these dependency tree graphs, we create two sub-tasks that differ in the direction of ground-truth graph traversal:

**WORLDOFBOXES-R2L.** Based on a randomly sampled tree, a *Root2Leaf* (WOB-R2L) question asks for the weight of a specific leaf box (target box) given only the weight of the root box (source box). The graph descriptions (e.g., box BGA weighs 2 pounds less than box LWR) and question (What is the weight of box EYU) shown on the left of Fig. 1 form a prompt in WOB-R2L.

**WORLDOFBOXES-L2R.** Based on a randomly sampled tree, a *Leaf2Root* (WOB-L2R) question asks for the weight of the root box (target box) given only the weight of a leaf box (source box). Based on the dependency tree graph shown in Fig. 1, a valid L2R question could ask for the weight of the root box RYH given the weight of any one of the leaf box (e.g., ABL or EYU).

### 3.2 SYNTHESIZING COT RATIONALES

To answer a question, a CoT must (1) find a path connecting the source and target box and (2) solve the weights of all boxes on the path. We construct four types of CoT rationale with different graph traversal strategies: (1) FORWARD-COT that only solve the boxes along the ground-truth path connecting the source to the target box, (2) BACKTRACK-COT that first backtracks from the target box to the source box, (3) RS-COT that traverses the entire dependency graph in random order, and (4) DFS-COT that traverses the graph following a depth-first-search procedure. We discuss each type CoT in details in Appendix A.1 and show example CoTs in Table 2.

## 4 DIAGNOSING IMPLICIT THINKING IN COT LEARNING

After introducing the WORLDOFBOXES dataset with different types of CoT rationales, we now turn to examine how training on these CoTs shapes the reasoning ability in LLMs. To this end, we evaluate post-SFT models on ID and OOD datasets and use linear probing to investigate how implicit thinking[4] emerges in these models. We show that the choice of CoT supervision systematically influences implicit reasoning in LLMs these hidden computations hamper models' generalization to more complex questions. Our experimental setup is explained in Appendix B.1.

---

[4]We provide a working definition of implicit thinking in Definition. 1.

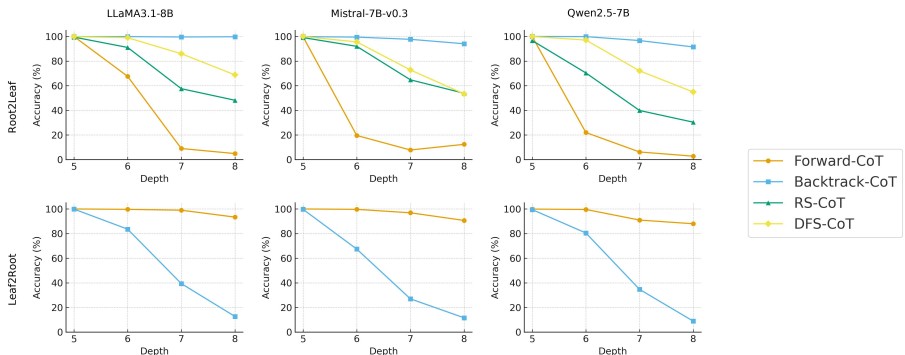

Figure 2: Test accuracy on WORLDOFBOXES of models trained with FORWARD-COT, BACKTRACK-COT, RS-COT, and DFS-COT. All models are trained on 100k questions with maximum dependency graph depth of 5 and evaluated on questions with depth 5 to 8. On WOB-L2R, both RS-COT and DFS-COT reduce to FORWARD-COT because there is only one admissible next box (the predecessor of the current box). Full results are shown in Table 3 and Table 4.

## 4.1 LENGTH GENERALIZATION RESULTS ON WORLDOFBOXES

For each WORLDOFBOXES task (R2L and L2R), we first finetune the base models on 100k questions based on dependency graphs up to 5 layers and one of the four types of CoT (FORWARD-COT, BACKTRACK-COT, RS-COT, DFS-COT). We then evaluate them on the corresponding ID and OOD test questions based on dependency graphs with up to 8 layers.

**Observation**: **BACKTRACK-COT generalizes in `R2L` while FORWARD-COT generalizes in `L2R`**. As shown in Fig. 2, all finetuned models show strong in-distribution (ID) questions based on dependency graphs of 5 layers only. However, on WORLDOFBOXES-R2L task, models trained with FORWARD-COT fail to generalize to out-of-distribution (OOD) questions with deeper dependency graphs than those seen in training: their performance drops to <10% on questions with 8-layer dependency graphs. In contrast, models trained with BACKTRACK-COT generalize to OOD questions significantly better, achieving at least 91% accuracy on questions with 8-layer dependency graphs.

On WORLDOFBOXES-L2R, we observe a reversed trend: models trained with FORWARD-COT generalize to more complex questions significantly better than models trained with BACKTRACK-COT. For example, LLaMA3.1-8B finetuned with FORWARD-COT only suffers a minimal performance drop (100%→94.5%) when the number of CoT steps increases from 5 to 8, while the same model trained with BACKTRACK-COT only achieves 12.7% accuracy on questions requiring 8 steps.

**Observation**: **DFS-COT generalizes better than RS-COT on WORLDOFBOXES-R2L**. We then finetune the three base models on the same WORLDOFBOXES-R2L training set used above, but with RS-COT and DFS-COT as the output supervision. Compared to FORWARD-COT and BACKTRACK-COT, these two types of CoT allow the model to explore the entire dependency graph following the order defined by a graph traversing algorithm and solving the weight of visited boxes along the way. As shown in Fig. 2, models trained with DFS-COT outperform their corresponding models trained with RS-COT in every OOD test (with dependency graphs of 6, 7, or 8 layers), while both obtain a significant advantage over FORWARD-COT: LLaMA3.1-8B's performance improves from 5.3% to 44.4% after we replaced the FORWARD-COT with DFS-COT as the training labels.

**Finding I**: **Reasoning in the direction with lower branching factor yields stronger generalization**. Reflecting upon the two observations above, we find that the length-generalization potential of CoT supervision depends on the interaction between CoT's traversal direction and the dependency graph's topology. Specifically, when the CoT traverses the dependency tree graph from leaf to root (e.g., BACKTRACK-COT in R2L), the post-SFT models generalize well to solve questions with dependency graphs deeper than those seen in training. This trend also aligns with the finding in (Zhang et al., 2025a), which states that given a question, it is easier to approach it from the direction with lower branching factor (e.g., traversing a tree from leaf to root has a branching factor of 1 at each step while traversing from root to leaf has a dynamic branching factor that equals the number of successors of the current node). In real-world reasoning scenarios, the underlying dependency

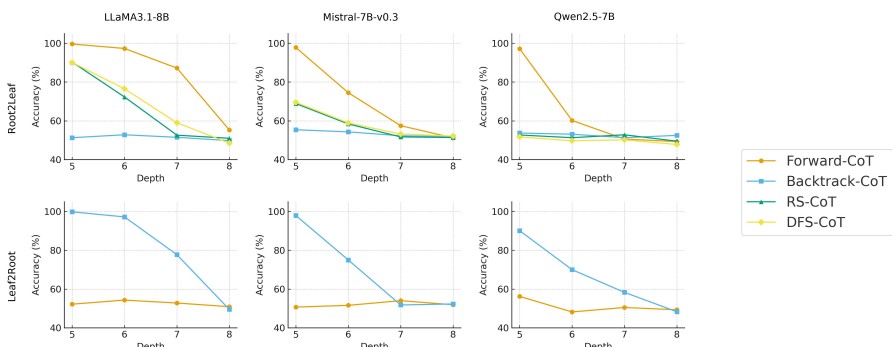

Figure 3: Probing accuracy of predicting whether a box described in the instruction is necessary (box PRU in Fig. 1) or not (box `CRU`/`FEB`/`JZF` in Fig. 1) in computing the target box. All models are trained on 100k questions with at most 5 layers in their dependency graph and then frozen. All linear probes are trained and evaluated on questions with dependency graphs of 5-8 layers. Full results are shown in Table 5 and Table 6.

graph is rarely as simple as a tree: some nodes could have multiple predecessors and successors and hence it is difficult to predict the optimal CoT direction. Therefore we need CoTs with a more general, error-tolerant traversal strategy like RS-CoT and DFS-CoT that can generalize well even when traversing the graph in the direction of large branching factor. Next, in Sec. 4.2, we will probe into the internal representations of models that learned different types of CoT and explain what factors enable/disable models' generalization.

## 4.2 Measuring Implicit Thinking with Linear Probe

In order to understand how the fine-tuned model internally decides which box to explore at a certain CoT step, we train a linear probe on the model's internal state for the binary classification task: whether a box is necessary for computing the weight of the target box (probing task **nece**($A$) in (Ye et al., 2024)). Further details of probing experiment's setup is discussed in Appendix C.1.

**Observation**: **Forward-CoT models implicitly think ahead on WoB-R2L while Backtrack-CoT models implicitly think ahead on WoB-L2R**. We present our probing results in Fig. 3. Similar to the findings in Ye et al. (2024), probing on WorldofBoxes-R2L requiring 5 CoT steps indicates that, by the end of the problem description and before generating the first token in CoT, the Forward-CoT model has already identified the full list of necessary boxes (the shortest path connecting the source root node to the target leaf node). This reveals that the generated CoT is simply following the result of the models' implicit thinking. In contrast to the results on WorldofBoxes-R2L, probing on the the L2R subtask, where the Forward-CoT models achieve significantly better generalization, shows that they do not learn to implicitly plan ahead: for both ID questions requiring 5 CoT steps and OOD questions requiring up to 8 steps, the probe's accuracy remains around random chance (50%). The probing results of the Backtrack-CoT models are exactly opposite of the Forward-CoT models: as shown in Fig. 3, the probe's accuracy remains around random chance at ID questions (depth=5) on WorldofBoxes-R2L while achieving > 90% accuracy on L2R.

**Observation**: **The implicit thinking ability cannot generalize to OOD questions**. While we observe hints of strong implicit thinking ability from probing Forward-CoT models and Backtrack-CoT models on R2L and L2R subtasks respectively, the probe's accuracy drops significantly on OOD questions with deeper dependency graphs. As shown in Fig. 3, the probing accuracy of all 3 Forward-CoT models (orange lines) on WoB-R2L (upper half of the figure) drops to below 60% on questions with dependency graphs of 8 layers. It indicates that implicit thinking can only support questions of the same or lower complexity as those seen during training. This drop in implicit thinking ability also aligns the same trend as their degrading performance shown in Fig. 2.

**Finding II**: **Implicit thinking negatively impacts length generalization**. Compared to the results of probing Forward-CoT models on R2L subtask, we observe lower accuracy when probing models trained on search traces (RS-CoT or DFS-CoT) on in-distribution questions of depth

5. For example, the probe only achieves an accuracy of 62% on Qwen2.5-7B-Rs-CoT and 51% on Qwen2.5-7B-DFS-CoT. Overall, probing these models trained with different types of CoT on WORLDOFBOXES reveals a negative correlation between their generalization to more complex questions and their implicit thinking ability: models that do not implicitly plan ahead (FORWARD-CoT on L2R, BACKTRACK-CoT and DFS-CoT on R2L) achieve better generalization to questions requiring more CoT steps than those seen in training. In other words, it suggests that implicit reasoning is a static capability with a fixed capacity, while explicit reasoning in the token space can dynamically extend its capacity beyond the training distribution. In the next section, we will propose a hypothesis based on information theory to explain what causes the models to/to not acquire the implicit reasoning ability on WORLDOFBOXES tasks.

# 5 TOWARDS GENERALIZABLE CoT BY ENCOURAGING EXPLICIT THINKING

In this section, we first propose a hypothesis regarding the "information gap" in CoT (Sec. 5.1) and show that it positively correlates with the implicit thinking ability of post-SFT models (Sec. 5.2). Finally, we show that a recent improvement to the SFT objective can reduce the implicit thinking of models trained on WORLDOFBOXES (Sec. 5.3).

## 5.1 THE HYPOTHESIS OF INFORMATION GAP IN CHAIN-OF-THOUGHT RATIONALE

**Notations.** We denote a directed acyclic dependency graph with $N$ nodes as $g = \{n_1, n_2, ..., n_N\}$. A trajectory $\tau = [\tau_1, \tau_2, ..., \tau_P]$ traverses $g$ by visiting nodes $\tau_1, \tau_2, ..., \tau_P$ sequentially. A trajectory $^{\phi}\tau$ follows a graph traversing heuristic $\phi$ (e.g., depth-first search) that cannot jump over an unvisited node to directly visit its children. For non-deterministic heuristic $\phi$ (e.g., DFS used in creating DFS-CoT), $^{\phi}\mathcal{T} = \{^{\phi}\tau\}$ is the collection of all possible trajectories $\tau$ that could be produced by $\phi$. At any step $i$, $\mathbb{C}_i^{\phi}$ is the list of *admissible nodes* that can be visited at step $i$ according to $\phi$. For the example in Fig. 1, $\mathbb{C}_1^{DFS} = \mathbb{C}_1^{RS} = \{RYH\}$ (because the first step has to visit the root) and $\mathbb{C}_2^{DFS} = \mathbb{C}_2^{RS} = \{CRU, FEB, JZF, PRU\}$. Assuming $\tau_2 = FEB$, then $\mathbb{C}_3^{DFS} = \{CPU, JRF\}$ because DFS must prioritize successors of the last visited node, while $\mathbb{C}_3^{RS} = \{CRU, JZF, PRU, CPU, JRF\}$. We first formally define implicit thinking on a dependency graph:

**Definition 1 (Implicit thinking on a dependency graph.)** *Given a dependency graph $g$ and the task of finding a path $\tau = [\tau_1, \tau_2, ..., \tau_P]$, we say the model "implicitly thinks" if the representations $r(\tau_i|g)$ and $r(c|g) \ \forall \tau_i \in \tau, c \in \mathbb{C}_i \setminus \tau$ are linearly separable.*

Note that the representations $r(\tau_i|g)$ and $r(c|g)$ are only conditioned on the graph without any generated CoT tokens. Therefore, Definition. 1 aligns with our approach to use a linear probe to quantify the magnitude of "implicit thinking" (i.e., the linear separability) within models. Next, we define the Information Gap within a CoT trajectory that traverses the dependency graph:

**Definition 2 (Information Gap within a CoT trajectory)** *Given a dependency graph $g$, the information gap $\mathcal{I}$ of a trajectory $\tau$ following a graph traversal heuristic $\phi$ is defined as:*

$$\mathcal{I}(\tau) = -\frac{1}{P} \sum_{i=1,...,P} \log(q^{\phi}(\tau_i|\tau)), \quad q^{\phi}(\tau_i|\tau) = \frac{|\mathbb{C}_i^{\phi} \cap \tau|}{|\mathbb{C}_i^{\phi}|}$$

where $P$ is the number of nodes visited by $\tau$ and $q^{\phi}(\tau_i|\tau)$ is the ratio between "explored admissible nodes" ($\mathbb{C}_i^{\phi} \cap \tau$) and "all admissible nodes" ($\mathbb{C}_i^{\phi}$) at step $i$. Intuitively, Definition. 2 measures both the "branching factor" and the "completeness" of the exploration at any point of the trajectory. A large branching factor corresponds to a large set of admissible nodes $\mathbb{C}_i$ and hence contributes to a larger denominator for $q^{\phi}(\tau_i|\tau)$. On the other hand, a more complete exploration of the dependency graph contributes to a larger numerator. For example in Fig. 1, if all of the four admissible boxes at layer 1 (CRU, FEB, JZF, PRU) are visited by the end of the DFS-CoT (i.e., $\mathbb{C}_2^{\phi} \in \tau$), then $|\mathbb{C}_2^{\phi} \cap \tau| = |\mathbb{C}_2^{\phi}|$ and hence $q^{dfs}(\tau_2|\tau) = 1$, yields an information gap of 0 at step 2. However, if only CRU and FEB are explored through the DFS-CoT, then $|\mathbb{C}_2^{\phi} \cap \tau| = 2$ at step 2, which would contribute to a positive information gap of $-\log(1/2)$.

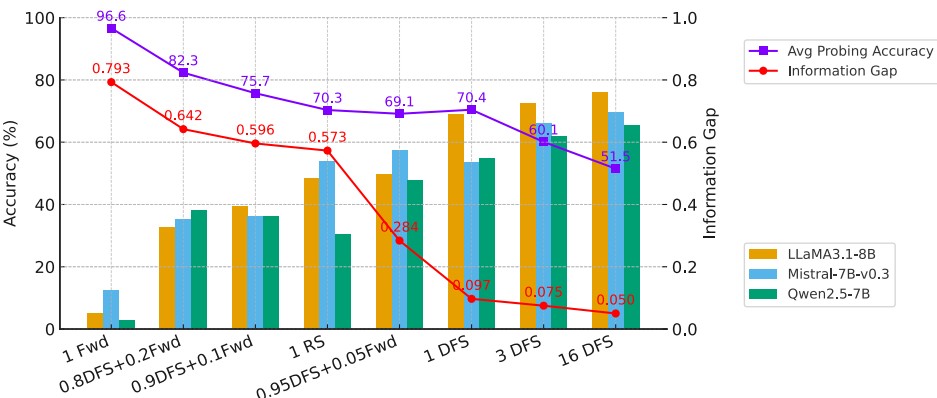

Figure 4: Answer accuracy on questions with 8-layer dependency graph, information gap of the training set, and probing accuracy averaged over the 3 base models. Full results are shown in Table 7.

**Hypothesis 1 (Information Gap in CoT induces implicit thinking)** *Given a language model $\mathcal{M}$, a dataset $\mathcal{G} = [(g_j, {}^{\phi_i}_j\tau)]_{j=1,\ldots,J}$ of $J$ dependency graphs and trajectories sampled from a traversal heuristic $\phi_i$. SFT $\mathcal{M}$ on $\mathcal{G}$ yields $\mathcal{M}_{\phi_i}$. If $\forall g_j \in \mathcal{G}$, $\mathcal{I}({}^{\phi_1}_j\tau) < \mathcal{I}({}^{\phi_2}_j\tau)$, then $\mathcal{M}^{\phi_1}$ has less implicit thinking (measured by probing) than $\mathcal{M}_{\phi_2}$.*

Intuitively, a large information gap between two consecutive CoT steps $i-1, i$ indicates that, given the previous $i-1$ steps, there are many possible nodes which can be explored at step $i$ and the current CoT $\tau$ only explores a few by the end (i.e., $|\mathbb{C}_i^\phi \cap \tau| << |\mathbb{C}_i^\phi|$). Therefore, simply optimizing the likelihood of explored nodes $|\mathbb{C}_i^\phi \cap \tau|$ could encourage the model to search for clues from its internal representations, which are not explicitly stated in previous CoT steps, that could be utilized to justify not exploring other admissible nodes.

## 5.2 SUPPORTING EVIDENCE OF THE INFORMATION GAP HYPOTHESIS

To support Hypothesis. 1, we present the next observation and finding that, although cannot establish causality, exemplify the co-occurrence of the mitigating Information Gap in the CoT training data and the reduction of the implicit thinking measured post-SFT.

**Observation**: **SFT on multiple search traces per question reduces implicit thinking and improves generalization**. In Definition. 2, we defined Information Gap within a single trajectory that traverses a dependency graph. When training an LLM to solve the target variable by traversing a dependency graph, it is possible to sample multiple CoT trajectories as the labels for each question. To gain more insights into the effect of learning from multiple CoTs per question, we define:

**Definition 3 (Information Gap within multiple CoT trajectories of a single graph)** *Given a dependency graph $g$ with a list of $K$ trajectories $\{\tau^k\}_{k=1,\ldots,K}$ that follow a graph traversal heuristic $\phi$, the information gap $\mathcal{I}$ within one trajectory $\tau^k$ is:*

$$\mathcal{I}(\tau^k) = -\frac{1}{P} \sum_{i=1,\ldots,P} \log(q^\phi(\tau_i^k|\mathcal{T})), \quad q^\phi(\tau_i|\mathcal{T}) = \frac{|\mathbb{C}_i^\phi \cap \mathcal{T}|}{|\mathbb{C}_i^\phi|}, \quad \mathcal{T} = \bigcup_k \tau^k$$

where $\tau_i^k$ is the $i$-th node's id visited by $\tau^k$. According to Definition. 3, given multiple trajectories of the same dependency graph, the information gap within each trajectory $\tau^k$ could be reduced because when exploring $\tau_i^k$, nodes visited by other trajectories ($\mathcal{T}$) that are also within the admissible nodes of the current step ($\mathbb{C}_i$) can increase the numerator ($|\mathbb{C}_i^\phi \cap \mathcal{T}|$). Therefore, Hypothesis. 1 suggests that once a model is shown multiple CoT trajectories that sufficiently explore the admissible node space at a state $g, \tau_{<i}$, it is less likely to develop implicit thinking.

To verify this hypothesis, we sample up to 16 DFS-CoT trajectories per question and then then finetune base models on this augmented training set with 3 CoTs per question. Based on the answer

| Training Loss | Math Accuracy | | | | Probe Accuracy |
|---|---|---|---|---|---|
| | 5 | 6 | 7 | 8 | 5 |
| SFT | 96.7 | 70.4 | 39.9 | 30.4 | 52.7 |
| DFT | 98.2 | 71.5 | 43.1 | 36.7 | 50.2 |

Table 1: Test accuracy on WORLDOFBOXES-R2L of Qwen2.5-7B trained with RS-COT using either standard SFT loss (cross entropy) or DFT loss (Wu et al., 2025). All models are trained on 100k questions with maximum dependency graph depth of 5 and evaluated on questions with dependency graphs of depth 5 to 8.

and probing accuracy in Fig. 4, LLaMA and Mistral models trained with 3/16 trajectories per question consistently obtain better length-generalization results and lower probing accuracy compared to their counterparts trained with only 1 trajectory per question. This observation corroborates our argument that mitigating the information gap by including multiple trajectories per question indeed reduces implicit thinking and hence improves length-generalization.

We calculate the average Information Gap of FORWARD-COT, RS-COT, and DFS-COT rationales of the entire training set. We also create 3 hybrid CoT types $x$DFS+$(1 - x)$FWD-CoT: at each step, we randomly select between DFS-COT and FORWARD-COT with probability $x$ and $1 - x$ respectively. As presented in Fig. 4, DFS-COT has the lowest average Information Gap of 0.097, while FORWARD-COT has the largest Information Gap of 0.793. For the augmented training set, with 3 DFS-COT trajectories per question the average information gap reduces from 0.097 to 0.075, while having 16 trajectories further lowers it down to 0.05. These measured information gaps ($\mathcal{I}$) directly validate Corollary. 3. Combining $\mathcal{I}$ and the probing accuracy reported in Fig. 4 further support our Hypothesis. 1, that information gap ($\mathcal{I}$) within CoT rationales induces implicit thinking in post-SFT models. Drawing connection between Hypothesis. 1 and Finding II that showcase how implicit thinking negatively impacts generalization further brings out our ultimate hypothesis:

**Hypothesis 2 (Information Gap in CoT impair length generalization)** *Given a language model $\mathcal{M}$, a dataset $\mathcal{G} = [(g_j, {}_j^\phi\tau)]_{j=1,...,J}$ of $J$ dependency graphs with maximum depth of $d$ and traversal trajectories following traversal heuristic $\phi$. SFT $\mathcal{M}$ on $\mathcal{G}$ yields $\mathcal{M}_\phi$. If $\forall g_j \in \mathcal{G}$, $\mathcal{I}({}_j^{\phi_1}\tau) < \mathcal{I}({}_j^{\phi_2}\tau)$, then $\mathcal{M}_{\phi_1}$ can generalize better than $\mathcal{M}_{\phi_2}$ on deeper dependency graphs $g'$ with depth $d' > d$.*

This hypothesis can be seen as the generalization of the conclusion in Zhang et al. (2025a), which states that among forward and backward reasoning, decoding in the direction with lower branching factor yields better results. By defining information gap based on the branching factor and exploration ratio, we are able to measure any traversal strategy (beyond forward and backward reasoning) and reveal that information gap in CoT supervision hurts generalization because it elicits static, implicit thinking that cannot generalize beyond the dependency graph depth seen in training.

## 5.3 CONNECTION TO REASONING TASKS IN THE WILD

In Definition. 2, we assume the dependency graph of variables is known. Here we define Information Gap within a CoT rationale for questions in the wild without a known dependency graph.

**Definition 4 (Information Gap within a CoT in the wild)** *Given an instruction $x$, the information gap $\mathcal{I}$ of a CoT rationale $y$ is the average token-level log likelihood:*

$$\mathcal{I}(y) = -\frac{1}{P} \sum_{i=1,...,P} \log(p_\theta(y_i|y_{<i}, x))$$

where $p_\theta(y_i|y_{<i}, x)$ is an LLM's (parameterized by $\theta$) output distribution given the previous $i - 1$ CoT tokens and instruction $x$. A large information gap is caused by low-likelihood tokens in CoT, which intuitively means a large portion of token-level probability space is left unexplored.

Since it's not a trivial task to establish a sequence of necessary variables for non-synthetic reasoning questions, we cannot measure the amount of implicit thinking in the wild by probing for these variables in models' internal representations. However, we observe that a recently proposed

SFT objective called DFT (Wu et al., 2025), which bring non-trivial gains in reasoning-heavy tasks like math and coding, can also reduce implicit thinking on WORLDOFBOXES. Specifically, DFT reweighs the token-level cross-entropy loss by the token's own probability. According to Definition. 4, low-likelihood tokens also contribute the most toward the overall information gap of the CoT rationale. Therefore, scaling down the loss of these tokens is also mitigating the influence of information gap in the gradients. We replace the cross-entropy loss with DFT loss in SFT and then train a linear probe following the same procedure in Sec. 4.2. Using RS-COT as the supervision in SFT, we observe stronger generalization to deeper graph and lower probing accuracy on models trained with DFT loss (Table 1), indicating that suppressing the gradients from CoT tokens with large information gap can indeed prevent models from developing implicit thinking during SFT.

## 6 CONCLUSION

In summary, we show that standard CoT training can mask non-causal "implicit" reasoning that collapses under length extrapolation, and we make this failure mode concrete with a controlled grade-school math benchmark (WORLDOFBOXES). SFT'd models solve in-distribution questions yet rely on implicit backtracking that does not generalize to deeper trees, as revealed by linear probes that recover the model's internal plan before any CoT token is emitted. By contrast, training models by adopting objectives that close *information gaps* (e.g., including multiple CoTs per question) suppresses these shortcuts in training, aligns generated CoT with the model's real computation, and yields markedly stronger out-of-distribution performance. Taken together, our results argue that length-generalizable reasoning emerges when supervision faithfully traces the causal steps the model actually uses, not when CoT is treated as a decorative afterthought.

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

# A  APPENDIX: SYNTHETIC MATH DATASET

## A.1  FOUR TYPES OF CoT RATIONALE

**Forward-CoT.** This type of CoT restricts reasoning to the boxes along the ground-truth path from the source to the target node. It forms the shortest sequence of boxes the model needs to solve in order to solve the target box. The CoT starts from the source box (the root for R2L or a leaf for L2R) and, at each step, performs an arithmetic operation to calculate the weight of a successor box.

**Backtrack-CoT.** This type of CoT differs from a FORWARD-CoT in that it first goes through the shortest path in the backward direction (starting from the target box and reaching the source box at the end) and then follows the exact steps in a FORWARD-CoT. BACKTRACK-CoT mimics the process of backtracking the dependencies from the knowledge graph (e.g., "I need to find the weight of Box X. Box X weighs 5 pounds more than Box H, so let me solve Box H first."), which is often adopted by humans in solving complex questions (Ye et al., 2024).

**Random-search (RS) CoT.** For R2L, we also create Random-search (RS-CoT) and DFS-CoT (introduced below), both of which can include boxes that are not necessary in solving the target box. Specifically, for each CoT step, we uniformly sample from a list of solvable boxes[5] and add the corresponding arithmetic solution for this box to the CoT.

**DFS-CoT.** Depth-first-search (DFS) CoT differs from RS-CoT in that it traverses the dependency graph following a DFS procedure. When the solved box at the current step has multiple children boxes, we randomly select one of them to solve at the next step.[6] Compared to random search, DFS has a smaller search space at most CoT steps.

## A.2  TRAINING AND TEST SETS

In constructing the training set, we first sample dependency tree graphs with the depth ranging from 3 to 5 and then create the prompt that describes all dependency between boxes. Fig. 1 shows a dependency graph of depth 5 and the partial prompts of a WOB-R2L question. Our in-distribution (ID) test set is created following the same procedure as the training set, with a fixed tree depth of 5. The random process in creating the dependency graph, box names, and box weights ensures that every test question describes a novel world of boxes. Other than the ID test set, we also create 3 out-of-distribution (OOD) test sets by sampling dependency tree graphs of depth 6, 7, and 8 so that solving these questions requires generating CoT longer than those seen in training.

---

[5]In R2L, a box is solvable if the weight of its parent is known. In L2R, a box is solvable if the weight of one of its children is known.

[6]This is achieved by randomizing the order of adding the children nodes to the stack during DFS.

| Type | CoT Rationales |
|---|---|
| FORWARD-CoT | Each Box RYH weighs 22 pounds. Box PRU weighs 1 pounds less than Box RYH. So Box PRU weighs 22 - 1 = 21 pounds. Box BXB weighs 6 pounds less than Box PRU. So Box BXB weighs 21 - 6 = 15 pounds. **[1 step omitted]** Box EYU weighs 4 pounds more than Box OGS. So Box EYU weighs 6 + 4 = 10 pounds. |
| BACKTRACK-CoT | Box EYU weighs 4 pounds more than Box OGS. So I need to find out the weight of Box OGS. Box OGS weighs 9 pounds less than Box BXB. So I need to find out the weight of Box BXB. **[1 step omitted]** Box PRU weighs 1 pounds less than Box RYH. So I need to find out the weight of Box RYH. *Now let's solve these unknown boxes one by one.* **[FORWARD-CoT omitted]** |
| RS-CoT | Each Box RYH weighs 22 pounds. Box FEB weighs 22 pounds less than Box RYH. So Box FEB weighs 22 - 22 = 0 pounds. Box CPU weighs 8 pounds more than Box FEB. So Box CPU weighs 0 + 8 = 8 pounds. Box CRU weighs 3 pounds less than Box RYH. So Box CRU weighs 22 - 3 = 19 pounds. **[20 steps omitted]** Box EYU weighs 4 pounds more than Box OGS. So Box EYU weighs 6 + 4 = 10 pounds. |
| DFS-CoT | Each Box RYH weighs 22 pounds. Box FEB weighs 22 pounds less than Box RYH. So Box FEB weighs 22 - 22 = 0 pounds. Box CPU weighs 8 pounds more than Box FEB. So Box CPU weighs 0 + 8 = 8 pounds. Box KRJ weighs 8 pounds less than Box CPU. So Box KRJ weighs 8 - 8 = 0 pounds. Box CSC weighs 7 pounds more than Box KRJ. So Box KRJ weighs 0 + 7 = 7 pounds. **[18 steps omitted]** Box EYU weighs 4 pounds more than Box OGS. So Box EYU weighs 6 + 4 = 10 pounds. |

Table 2: Four types of CoT rationales we created for the WORLDOFBOXES-R2L question shown in Fig. 1: "what is the weight of Box EYU?". BACKTRACK-CoT first backtracks from the target box EYU to the source box RYH and then produces the FORWARD-CoT (omitted). RS-CoT traverses the tree by randomly choosing a solvable boxes whose predecessor's weight is known. DFS-CoT traverses the tree graph following a depth-first search heuristic until it reaches the target box EYU. We omit some intermediate steps in each CoT rationale due to limited space.

| Models | ROOT2LEAF (R2L) | | | | LEAF2ROOT (L2R) | | | |
|---|---|---|---|---|---|---|---|---|
| | 5 | 6 | 7 | 8 | 5 | 6 | 7 | 8 |
| FORWARD-COT | | | | | | | | |
| LLaMA3.1-8B | 100 | 67.6 | 9.0 | 4.9 | 100 | 99.7 | 99 | 93.4 |
| Mistral-7B-v0.3 | 99.7 | 19.6 | 7.8 | 12.4 | 100 | 99.7 | 96.9 | 90.7 |
| Qwen2.5-7B | 99.9 | 22.0 | 6.2 | 2.8 | 99.9 | 99.6 | 91.0 | 88.0 |
| BACKTRACK-COT | | | | | | | | |
| LLaMA3.1-8B | 99.9 | 99.9 | 99.6 | 99.8 | 99.9 | 83.6 | 39.4 | 12.7 |
| Mistral-7B-v0.3 | 99.9 | 99.5 | 97.7 | 94.1 | 99.7 | 67.4 | 27 | 11.6 |
| Qwen2.5-7B | 100 | 99.9 | 96.7 | 91.5 | 99.5 | 80.5 | 34.7 | 8.9 |

Table 3: Test accuracy on WORLDOFBOXES of models trained with FORWARD-COT and BACKTRACK-COT. All models are trained on 100k questions with maximum dependency graph depth of 5 and evaluated on questions with dependency graphs of depth 5 to 8.

| Models | RS-COT | | | | DFS-COT | | | |
|---|---|---|---|---|---|---|---|---|
| | 5 | 6 | 7 | 8 | 5 | 6 | 7 | 8 |
| LLaMA3.1-8B | 99.5 | 91.1 | 57.6 | 48.2 | 99.9 | 99.0 | 86.0 | 68.8 |
| Mistral-7B-v0.3 | 99.1 | 92.0 | 64.8 | 53.9 | 100.0 | 95.6 | 72.8 | 53.4 |
| Qwen2.5-7B | 96.7 | 70.4 | 39.9 | 30.4 | 100.0 | 97.1 | 72.1 | 54.9 |

Table 4: Test accuracy on WORLDOFBOXES-R2L of models trained with RS-COT and DFS-COT. All models are trained on 100k questions with maximum dependency graph depth of 5 and evaluated on questions with dependency graphs of depth 5 to 8.

# B APPENDIX: EXPERIMENTS

## B.1 EXPERIMENTAL SETUP

**Base models.** For all experiments in this work, we finetune and evaluate on three base models: LLaMA3.1-8B (Dubey et al., 2024), Mistral-7B-v0.3 (Jiang et al., 2023), and Qwen2.5-7B (Qwen et al., 2025). During evaluation, we allow the models to generate up to 16384 tokens to minimize the risk of failing the task by running out of token budget.

## B.2 FULL RESULTS

We show the full results of evaluating post-SFT models on WORLDOFBOXES test sets in Table 3 and Table 4.

## C  APPENDIX: PROBING EXPERIMENTS

### C.1  EXPERIMENTAL SETUP

In order to understand how the fine-tuned model internally decides which box to explore at a certain CoT step, we train a linear probe on the model's internal state for the binary classification task: whether a box is necessary for computing the weight of the target box (probing task $\mathbf{nece}(A)$ in (Ye et al., 2024)). Specifically, we append a box's name to the problem description and feed the last token's final-layer representation as the input to the probe. With all model parameters frozen, we train the linear probe on 100k WORLDOFBOXES (either R2L or L2R) questions requiring 3 to 8 CoT steps with a balanced class distribution.[7] The test set is comprised of unseen questions with a balanced distribution of positive and negative classes.

### C.2  MORE FINDINGS

**Observation**: **FORWARD-CoT models implicitly search backward while BACKTRACK-CoT models implicitly search forward**.   Other than the overall accuracy of probing for every necessary box in the $k$-step CoT, we further break down the probe's accuracy at classifying boxes at different depths in the dependency graph. On WOB-R2L questions with out-of-distribution depths of 6 to 8, the probes on FORWARD-CoT models consistently achieve strong performance in classifying boxes in the last 5 layers. This observation indicates that FORWARD-CoT models learn static, implicit backtracking: before generating the CoT, they internally search backward from the target box to uncover the last 5 necessary boxes. In contrast, on WOB-L2R, we can observe that the probes on BACKTRACK-CoT models remain accurate in the first 5 boxes from the leaf-to-root path, only failing to classify boxes in the last $k - 5$ layers (i.e., boxes that are closer to the root), suggesting BACKTRACK-CoT models learn static, implicit forward search that can discover the first 5 necessary boxes that need to be solved in CoT.

**Finding III**: **Implicit thinking traverse the dependency graph from leaf to root**.   Reflecting on the symmetrical pattern observed above: FORWARD-CoT models implicitly think backward on R2L while BACKTRACK-CoT models implicitly think forward on L2R, we find that models always develop implicit thinking when the CoT rationales present the path from root to leaf. Moreover, this implicit search always proceeds from bottom up (leaf to root), which has a branching factor of 1, instead of attempting the much more complex task of traversing the entire graph from top down (root to leaf). When the CoT rationales already traverse the tree graph from leaf to root, then the models would not develop additional implicit search. In real-world reasoning scenarios, the underlying dependency graph could encompass a mixture of R2L and L2R paths: some nodes may have more predecessors than successors and some may have more successors than predecessors. Therefore, a purely FORWARD-CoT or BACKTRACK-CoT may inevitably induce implicit thinking in the models.

## D  MORE RESULTS

### D.1  FULL RESULTS OF FIGURES IN THE MAIN PAPER

We show the full results of probing experiments in Table 5 and Table 7.

### D.2  IMPACT OF MODEL SIZES ON IMPLICIT THINKING

We also finetune Qwen2.5 models of different sizes (3B, 7B, and 14B) on our WOB training set with graph depth up to 5. We show the OOD evaluation accuracy and ID probing accuracy in Table 8. We observe that the larger model (14B) always achieve better accuracy on questions with graph depth up to 8 under every CoT type, demonstrating a positive scaling in OOD length generalization. When trained with FORWARD-CoT, the probing accuracy is lower on Qwen2.5-14B (83%) compared to

---

[7]For each question of depth $k$, we create $k - 1$ positive examples by appending each of the necessary boxes, except for the root box, to the question. We also create $k - 1$ negative examples by appending one random unnecessary sibling box of each necessary box.

| Models | ROOT2LEAF (R2L) | | | | LEAF2ROOT (L2R) | | | |
|---|---|---|---|---|---|---|---|---|
| | 5 | 6 | 7 | 8 | 5 | 6 | 7 | 8 |
| FORWARD-CoT | | | | | | | | |
| LLaMA3.1-8B | 99.6 | 97.3 | 87.2 | 55.3 | 52.2 | 54.3 | 52.8 | 50.9 |
| Mistral-7B-v0.3 | 97.8 | 74.5 | 57.5 | 51.2 | 50.7 | 51.6 | 54 | 51.9 |
| Qwen2.5-7B | 97.2 | 60.2 | 50.6 | 49.3 | 56.2 | 48.2 | 50.5 | 49.3 |
| BACKTRACK-CoT | | | | | | | | |
| LLaMA3.1-8B | 51.3 | 52.8 | 51.5 | 49.8 | 99.8 | 97.2 | 77.7 | 49.5 |
| Mistral-7B-v0.3 | 55.4 | 54.3 | 52.3 | 51.8 | 97.9 | 75.0 | 51.8 | 52.3 |
| Qwen2.5-7B | 53.7 | 53.1 | 51.3 | 52.5 | 90.1 | 70.0 | 58.3 | 48.3 |

Table 5: Probing accuracy of predicting whether a box described in the instruction is necessary (box PRU in Fig. 1) or not (box CRU/FEB/JZF in Fig. 1) in computing the target box. All models are trained on 100k questions with at most 5 layers in their dependency graph and then frozen. All linear probes are trained and evaluated on questions with dependency graphs of depth 5-8.

| Models | RS-CoT | | | | DFS-CoT | | | |
|---|---|---|---|---|---|---|---|---|
| | 5 | 6 | 7 | 8 | 5 | 6 | 7 | 8 |
| LLaMA3.1-8B | 90.3 | 72.3 | 52.6 | 51.0 | 90.1 | 76.5 | 59.0 | 48.5 |
| Mistral-7B-v0.3 | 68.9 | 58.4 | 51.7 | 51.4 | 69.5 | 58.9 | 53.2 | 52.1 |
| Qwen2.5-7B | 52.7 | 51.3 | 52.8 | 49.5 | 51.7 | 49.7 | 50.2 | 47.8 |

Table 6: Probing accuracy on WORLDOFBOXES-R2L of models trained with RS-CoT and DFS-CoT. All models are trained on 100k questions with maximum dependency graph depth of 5 and probed on questions with dependency graphs of depth 5 to 8.

smaller models while its answer accuracy on OOD questions with depth of 6 is significantly higher, suggesting that scaling up the model size could elicit more explicit reasoning.

| Models | Answer (acc.) | | | | Information Gap | Probing (acc.) |
|---|---|---|---|---|---|---|
| | 5 | 6 | 7 | 8 | 3-5 | 5 |
| 1 FORWARD-CoT | | | | | | |
| LLaMA3.1-8B | 100 | 67.6 | 9.0 | 4.9 | | 94.9 |
| Mistral-7B-v0.3 | 99.7 | 19.6 | 7.8 | 12.4 | 0.793 | 97.8 |
| Qwen2.5-7B | 99.9 | 22.0 | 6.2 | 2.8 | | 97.2 |
| 1 (80%DFS-CoT + 20%FORWARD-CoT) trajectory | | | | | | |
| LLaMA3.1-8B | 99.9 | 71.0 | 35.8 | 32.5 | | 87.8 |
| Mistral-7B-v0.3 | 100 | 70.8 | 42.8 | 35.3 | 0.642 | 85.5 |
| Qwen2.5-7B | 99.0 | 74.6 | 39.7 | 38.1 | | 73.7 |
| 1 (90%DFS-CoT + 10%FORWARD-CoT) trajectory | | | | | | |
| LLaMA3.1-8B | 99.7 | 72.2 | 41.5 | 39.4 | | 85.4 |
| Mistral-7B-v0.3 | 99.8 | 66.1 | 44.5 | 36.1 | 0.596 | 74.5 |
| Qwen2.5-7B | 98.6 | 75.9 | 45.5 | 36.1 | | 67.3 |
| 1 RS-CoT | | | | | | |
| LLaMA3.1-8B | 99.5 | 91.1 | 57.6 | 48.2 | | 90.3 |
| Mistral-7B-v0.3 | 99.1 | 92.0 | 64.8 | 53.9 | 0.573 | 68.0 |
| Qwen2.5-7B | 96.7 | 70.4 | 39.9 | 30.4 | | 52.7 |
| 1 (95%DFS-CoT + 5%FORWARD-CoT) trajectory | | | | | | |
| LLaMA3.1-8B | 99.9 | 81.8 | 53.2 | 49.5 | | 82.3 |
| Mistral-7B-v0.3 | 99.9 | 92.3 | 66.7 | 57.2 | 0.284 | 65.9 |
| Qwen2.5-7B | 98.8 | 91 | 57.9 | 47.7 | | 59.2 |
| 1 DFS-CoT trajectory | | | | | | |
| LLaMA3.1-8B | 99.9 | 99.0 | 86.0 | 68.8 | | 90.1 |
| Mistral-7B-v0.3 | 100.0 | 95.6 | 72.8 | 53.4 | 0.097 | 69.5 |
| Qwen2.5-7B | 100.0 | 97.1 | 72.1 | 54.9 | | 51.7 |
| 3 DFS-CoT trajectories | | | | | | |
| LLaMA3.1-8B | 100.0 | 99.6 | 88.2 | 72.4 | | 67.7 |
| Mistral-7B-v0.3 | 99.9 | 98.9 | 82.7 | 66.0 | 0.075 | 60.0 |
| Qwen2.5-7B | 100.0 | 97.1 | 74.7 | 61.9 | | 52.6 |
| 16 DFS-CoT trajectories | | | | | | |
| LLaMA3.1-8B | 100.0 | 99.8 | 90.4 | 76.0 | | 53.7 |
| Mistral-7B-v0.3 | 99.9 | 99.2 | 84.3 | 69.6 | 0.050 | 49.5 |
| Qwen2.5-7B | 100.0 | 98.6 | 76.9 | 65.3 | | 51.3 |

Table 7: Test and Probing accuracy on WORLDOFBOXES-R2L of models trained with FORWARD-CoT, RS-CoT, and multiple DFS-CoT trajectories. All models are trained on 100k questions requiring at most 5 steps in CoT and evaluated on questions requiring 5, 6, 7, and 8 steps in CoT.

| Models | Answer (acc.) | | | | Information Gap | Probing (acc.) |
|---|---|---|---|---|---|---|
| | 5 | 6 | 7 | 8 | 3-5 | 5 |
| 1 FORWARD-COT | | | | | | |
| Qwen2.5-3B | 99.7 | 11.6 | 2.9 | 4.6 | | 96.8 |
| Qwen2.5-7B | 99.9 | 22.0 | 6.2 | 2.8 | 0.793 | 97.2 |
| Qwen2.5-14B | 100 | 46.5 | 4.3 | 6.2 | | 83.0 |
| 1 RS-COT | | | | | | |
| Qwen2.5-3B | 92.3 | 63.1 | 31.3 | 27.8 | | 51.3 |
| Qwen2.5-7B | 96.7 | 70.4 | 39.9 | 30.4 | 0.573 | 52.7 |
| Qwen2.5-14B | 99.0 | 87.0 | 50.2 | 40.5 | | 51.4 |
| 1 DFS-COT | | | | | | |
| Qwen2.5-3B | 99.8 | 89.3 | 54.9 | 38.9 | | 51.4 |
| Qwen2.5-7B | 100 | 97.1 | 72.1 | 54.9 | 0.097 | 51.7 |
| Qwen2.5-14B | 100 | 98.6 | 81.5 | 69.8 | | 51.5 |

Table 8: Test and Probing accuracy on WORLDOFBOXES-R2L of Qwen2.5-3B/7B/14B trained with FORWARD-COT, RS-COT, and multiple DFS-COT trajectories. All models are trained on 100k questions requiring at most 5 steps in CoT and evaluated on questions requiring 5, 6, 7, and 8 steps in CoT.

# E   APPENDIX: MORE DETAILS OF THE INFORMATION GAP HYPOTHESIS

## E.1   COROLLARIES

Here, we present a few corollaries that follow Definition. 2.

**Corollary 1** $\forall(g, {}^{back}\tau) \in \text{WOB-}R2L$ *s.t.* ${}^{back}\tau$ *follows* BACKTRACK-COT, $\mathcal{I}({}^{back}\tau) = 0$.

**Corollary 2** $\forall(g, {}^{fwd}\tau) \in \text{WOB-}L2R$ *s.t.* ${}^{fwd}\tau$ *follows* FORWARD-COT, $\mathcal{I}({}^{fwd}\tau) = 0$.

Corollary. 1 holds because BACKTRACK-COT follows a leaf-to-root path on WOB-R2L and then revisits this discovered path from root to leaf to calculate the box weights; hence it has only one possible node to visit next, that is $\forall i \leq P$, $|\mathbb{C}_i^{back}| = 1$ and thus $q^{back}(\tau_i|\tau) = 1$. Corollary. 2 can be proved in a similar manner as, when traversing a leaf-to-root path on WOB-L2R, FORWARD-COT has only one possible visitable node at any step so that $\forall i \leq P$, $|\mathbb{C}_i^{fwd}| = 1$.

**Corollary 3** $\forall(g, {}^{dfs}\tau, {}^{rs}\tau) \in \text{WOB-}R2L$ *with a constant branching factor for every node, s.t.* ${}^{dfs}\tau$ *follows* DFS-COT *and* ${}^{rs}\tau$ *follows* RS-COT, $\mathcal{I}({}^{dfs}\tau) \leq \mathcal{I}({}^{rs}\tau)$.

We back up Corollary. 3 by empirical evidence: we measure the average information gap of DFS-COT and RS-COT over the entire training set and present the results in Fig. 4 and Table 7.

