# OpenReview forum: "Information Gap in Chain-of-Thought Induces Implicit Thinking that Fails in Length Generalization"
_ICLR.cc/2026/Conference — Submitted to ICLR 2026_

### Official Review · Reviewer_yrFH · 2025-10-29

**Soundness:** 4
**Presentation:** 4
**Contribution:** 4
**Rating:** 8
**Confidence:** 4

**Summary:**

This paper investigates a critical failure mode of Chain-of-Thought (CoT) reasoning: the model's inability to generalize to problems requiring longer reasoning chains than those seen during training (i.e., length generalization).

The authors introduce a controllable synthetic math dataset called WORLDOFBOXES (WOB) to diagnose this failure. They find that models trained with standard supervised finetuning (SFT) on CoT rationales do not learn to reason explicitly. Instead, they learn an "implicit thinking" shortcut, internally identifying all necessary variables before generating the first CoT token. This implicit ability is shown to be static and fails when the problem complexity (e.g., graph depth) exceeds the training distribution, causing a collapse in performance.

The paper hypothesizes that this shortcut is induced by a large "Information Gap" in the- training data—a significant disparity between the many "admissible" reasoning actions at each step and the single "golden" path provided in the CoT. This gap, they argue, forces the model to find an internal justification for the specific path, leading to this non-generalizable implicit shortcut.

To validate this hypothesis, the authors provide a series of strong pieces of evidence. First, they use linear probing  to find that the models failing at generalization are precisely the ones that exhibit high "implicit thinking" on in-distribution data; this implicit ability then collapses when faced with out-of-distribution (OOD) data. Second, they show a positive correlation  between the quantified size of the "Information Gap" for different CoT types (e.g., FORWARD-COT vs. DFS-COT) and the degree of "implicit thinking" they induce. Finally, and most critically, they demonstrate through an intervention experiment (Intervention) that actively reducing the Information Gap does indeed suppress implicit thinking and significantly improves the model's  generalization.

**Strengths:**

1. The paper's primary strength is its  experimental design. The WOB synthetic dataset allows the authors to control for all confounding variables (domain knowledge, problem structure, complexity) and cleanly measure the variables of interest: CoT type, implicit thinking, and generalization.
2. The "Information Gap" is a novel concept that provides a compelling explanation for why models take "implicit" shortcuts. The authors demonstrate it can be measured and, more importantly, mitigated to produce  improvements.
3. The paper  connects the model's internal representations (measured via probing) to its external behavior (task accuracy).
4. The paper is exceptionally clear. The logical flow from problem to diagnosis to hypothesis and solution is compelling and easy to follow.

**Weaknesses:**

The paper's weaknesses are minor and are largely limitations inherent to its (very strong) controlled methodology.

The core claims are validated exclusively on the synthetic WOB dataset. This control is a strength, but it leaves open the question of how this mechanism operates in "real-world" reasoning tasks (e.g., GSM8k, coding) which are messier and entangled with world knowledge.

**Questions:**

1. The connection to real-world tasks via DFT loss is intriguing. A more direct test would be to apply your core intervention (training on multiple CoT trajectories) to a real-world dataset. Have the authors considered creating or finding a dataset like GSM8k-multi-path, where the *same* math problem is solved using 2-3 different (but valid) reasoning chains? This would be a powerful demonstration that the Information Gap principle holds beyond WOB.
2. The experiments are run on 7B/8B models. Do the authors have any insights or preliminary results on how model scale affects this phenomenon? Does "implicit thinking" become more or less prevalent in much larger models (e.g., 70B+)? Or, is it a fundamental artifact of the SFT-on-CoT process, regardless of scale?

---

> ### Author Response · Authors · 2025-11-22
> **Response to Reviewer yrFH**
>
> We sincerely thank the reviewer for their kind comments and constructive suggestions for our work. We especially appreciate that you describe our Information Gap “as a novel concept that provides a compelling explanation for why models take "implicit" shortcuts" and our diagnosis experimental design as "clean, strong" and "compelling and easy to follow"; while admitting that "the main limitations are inherent to its (very strong) controlled methodology". We provide our answers to the reviewer's questions below:
>
> -- *“The core claims are validated exclusively on the synthetic WOB dataset. ..., Is there be a powerful demonstration that the Information Gap principle holds beyond WOB?”*
>
> > There is a recent study (Liu et al., 2025) from NVIDIA that shows that scaling the SFT dataset by increasing the number of CoT responses per prompt can effectively improving the performance of the reasoning model on AIME 25 and LiveCodeBench (Sec 4.4).  We also conduct a study where we (1) sample up to 16 CoT rationales from Qwen2.5-32B for each question in OpenAI MATH dataset (12k); (2) SFT a Qwen2.5-7B model using either 1, 6, or 16 CoTs per question; and (3) evaluate the finetuned model on the held-out MATH-500 test set. Results show that the model trained with 16 CoTs per question outperform the model trained with 6 or 1 CoTs per question (67% VS 62% VS 59%).
>
> > Liu, Zihan, et al. "AceReason-Nemotron 1.1: Advancing Math and Code Reasoning through SFT and RL Synergy." arXiv preprint arXiv:2506.13284 (2025).
>
> -- *“Does "implicit thinking" become more or less prevalent in much larger models"*
>
> > Unfortunately we do not have the resources to train models with 70B+ parameters efficiently. But we agree this is a very important insight and we have started training Qwen2.5-3B and Qwen2.5-14B models on our WoB datasets and will share the evaluation and probing results as soon as possible!

---

> > ### Author Response · Authors · 2025-12-02
> > **Results on smaller and larger base models**
> >
> > We updated the paper with results (both math and probing) from Qwen2.5-3B and Qwen2.5-14B in Appendix D.2 and Table 8. We observe a positive scaling in OOD length generalization and a negative scaling in implicit thinking (as measured by probing accuracy).

---

### Official Review · Reviewer_fiU3 · 2025-10-29

**Soundness:** 2
**Presentation:** 3
**Contribution:** 2
**Rating:** 4
**Confidence:** 3

**Summary:**

The authors investigate how supervised fine-tuning with different types of chain of thought traces affects performance on the WorldOfBoxes task, a simple GSM-type task involving object relationship and addition/subtraction of small integers.

They define implicit thinking based on whether the internal weights of the model can be used to predict which boxes are important to the task, as measured by linear probing. By applying SFT with different CoT techniques, they are able to elicit different amounts of implicit thinking. They find that implicit thinking negatively impacts length generalization. Further, they hypothesize a relationship between implicit thinking and the ‘information gap’ of the chain of thought traces.

**Strengths:**

The problem investigated in the paper is timely and well-motivated.

The World of Boxes problem is a clever way to isolate different types of reasoning traces. Since it requires only simple arithmetic, the main variable being tested is finding the correct L2R/R2L path.

The authors successfully connect SFT with different types of chain of thought traces to a concrete difference in implicit thinking as measured by linear probing. They also make this connection with length generalization, showing significant evidence that implicit thinking is associated with worse length generalization.

**Weaknesses:**

The paper could be improved by a formalization of ‘implicit thinking’. The authors measure it by using a linear probe to predict from the model’s weights whether a box is relevant to the WoB instance. If the probe is successful, they say that the model is performing implicit thinking. It would be nice to see an explicit definition like “model M is an implicit thinking model if (condition)”, or more likely “model M thinks implicitly on problem P if (condition)”.

The central hypothesis of the paper is that this specific computable quantity, the information gap in chain of thought, induces implicit thinking. However, the second half of the hypothesis, $M^{\phi_1}$ has less implicit thinking (as measured by probing) than $M^{\phi_2}$, is not specific enough to be falsifiable.

Further, I don’t get a strong sense for why the authors believe that the information gap is the relevant variable. The experimental evidence is limited (there are only five data points for the information gap in fig. 4, and the experiment only shows correlation). This could be strengthened via some mathematical intuition – not necessarily a formal proof, but maybe showing the relationship between information gap and implicit thinking in some simplified setting.

**Questions:**

I noticed that the probing accuracy for RS-COT and DFS-COT is the same despite a large difference in the information gap. Why do you think this is?

Do you think of implicit thinking as a binary state (i.e., either the model solves the problem by thinking implicitly or it solves the problem by step-by-step reasoning)? Is there evidence that models can use different degrees of implicit thinking?

Typos:
Sentence 1 of abstract: “Chain of Though(t)”

In p.7 Hypothesis 1, the trajectories are labelled as $^{\phi_1} \tau$ and $\tau^{\phi_1}$? Do these have different meanings, or is it a typo?

Additionally wrote both $M_{\phi_i}$ and $M^{\phi_i}$ in Hypothesis 1 and 2.

---

> ### Author Response · Authors · 2025-11-22
> **Response to Reviewer fiU3**
>
> We thank the reviewer for their constructive feedback. We especially appreciate that they describe our World of Boxes problem as *"a clever way to isolate different types of reasoning traces"* and acknowledged that we show *" significant evidence that implicit thinking is associated with worse length generalization"*. We provide our answers to reviewer's questions below:
>
> **Definition of Implicit Thinking**
>
> > We provide the following definition of implicit thinking and added to the paper:
> >
> > > Definition: Given a dependency graph $g$ with nodes $C$ and the task of finding a path $\tau=[\tau_1, \tau_2, ..., \tau_P]$, we say the model “implicitly thinks” if the representations of $\tau_i\in\tau$: $r(\tau_i|g)$ and $c \in C\setminus\tau$: $r(c|g)$ are linearly separable.
> >
> > Note that the representations $r(\tau_i|g)$ and $r(c|g)$ are only conditioned on the graph, without any generated chain-of-thoughts tokens. Given this definition, it is natural to use linear probe to quantify the magnitude of “implicit thinking” (i.e., the linear separability) within models. The hypothesis we are trying to test is that “the magnitude of implicit thinking is negatively correlated with the model’s ability to generalize to deeper graphs (line 325-327 in the first version)”.
>
> **Experimental evidence showing the importance of the Information Gap metric**
>
> >We reiterate our empirical findings and show some new results below to support our argument that Information Gap is a very relevant and useful metric for predicting the generalization ability of finetuned models.
>
> -- *"There are only five data points for the information gap in fig. 4."*
>
> > We extend Fig. 4 which now includes 8 data points. Specifically, we add 3 new hybrid CoT types, xDFS+(1-x)Fwd-CoT where x = 0.95, 0.9, and 0.8: when sampling the next node (box) from all admissible nodes, we directly select a node on the ground-truth ForwardCoT path with probability $(1-x)$ and follow DFS with probability $x$. Note that it degrades to ForwardCoT when x = 0 as it always select the ground-truth node. We evaluate models trained with these 3 hybrid CoT types on in-distribution and OOD test sets up to depth of 8 and added the results to Figure 4 and Table 6. Overall, they follow the same trend, where higher information gap leads to higher probing accuracy and lower OOD generalization to deeper graphs. The extended data points confirm our hypothesis regarding the strong negative correlation between Information Gap and length generalization ability.

---

> > ### Author Response · Authors · 2025-11-22
> > **Response to Reviewer fiU3 (cont.)**
> >
> > -- *“Mathematical intuition of how Information Gap in CoT induces implicit thinking”*
> >
> > > ### 1. Setup: two reasoning strategies on a tree
> > >
> > > Consider one WoB instance with a dependency tree of depth $k$. A CoT trajectory
> > >
> > > $$
> > > \tau = (\tau_1, \dots, \tau_P)
> > > $$
> > >
> > > visits nodes in the tree in some order. Let $C_i$ be the set of admissible nodes at step $i$ under a fixed traversal heuristic $\phi$ (i.e., $C_i = C_i^\phi$ in the notation of the paper). We consider two competing ways for the model to implement the mapping from (question, CoT prefix) to the next CoT token during training (with teacher forcing):
> > >
> >
> > > **A. Explicit / local exploration (no implicit planning).**
> > >    At step $i$, the model only uses the observable prefix $(\text{question}, \tau_{<i})$, without internally computing the ground-truth path. Because siblings in the tree are exchangeable (random labels, random weights), unexplored admissible nodes in $C_i$ are indistinguishable. The best such a model can do is approximate a uniform choice and the optimal per-step cross-entropy loss at step $i$ is:
> > >   $$
> > >   \ell_{\text{expl}}(i)
> > >   = -\log p_{\text{expl}}(\tau_i \mid \text{question}, \tau_{<i})
> > >   = -\log \frac{1}{\lvert C_i \rvert}
> > >   = \log \lvert C_i \rvert.
> > >   $$
> > >
> > > **B. Implicit planning (by backtracking) before CoT.**
> > >    Before emitting any CoT token, the model internally computes the entire path (e.g., by backtracking from the target to the source node). Then at step $i$, it already “knows” the correct next node on this path, so it achieves an optimal loss of 0:
> > >   $$
> > >   \ell_{\text{impl}}(i)
> > >   = -\log p_{\text{impl}}(\tau_i \mid \text{question}, \tau_{<i})
> > >   \approx 0.
> > >   $$
> > >  Because training minimizes token-level cross-entropy, the model is incentivized to adopt whichever strategy gives higher per-token log-probability.
> > >  Thus, whenever there is more than one unexplored admissible node (i.e., $\lvert C_i \rvert > 1$), the **explicit strategy** pays a **strictly positive penalty** $\log \lvert C_i \rvert$, while the implicit strategy can drive the loss toward 0.
> > >
> > > ---
> > >
> > > ### 2. Connecting to trajectory-level information gap
> > > Now recall the information gap defined in Definition 1:
> > > $$
> > > \mathcal{I}(\tau)
> > > = -\frac{1}{P}\sum_{i=1,...,P} \log\frac{|C_i \cap \tau|}{|C_i|}
> > > = \frac{1}{P}\sum_{i=1,...,P} \left(\log|C_i| - \log|C_i \cap \tau|\right).
> > > $$
> > >
> > > Here, the first term inside the summation $\log|C_i|$ is the **maximum explicit-exploration penalty** dictated purely by the branching structure (discussed above). The second term $\log|C_i \cap \tau|$ measures how many of those admissible options are actually **supported by the entire supervision** $\tau$. For example, $\tau_{i'}\in C_i$ is a negative node at step $i$ that contribute to the explicit-exploration penalty, so that gradients at step $i$ push its probability down. However, $\tau_{i'}$ appear later as the positive node at step $i'$ and thus the gradients incurred at step $i'$ push its probability up, which partially offset the incentive to drive its probability to zero at step $i$ using implicit thinking.
> > > The information gap $\mathcal{I}(\tau) = \sum_{i=1,...,P} \log|C_i| - \log|C_i \cap \tau|$
> > > is precisely the **residual ambiguity**: the portion of the explicit-exploration penalty that is *not* explained away by global, trajectory-level supervision and thus must be resolved by **implicit thinking** in the hidden state.

---

> > > ### Author Response · Authors · 2025-11-22
> > > **Response to Reviewer fiU3 (cont., Part 3)**
> > >
> > > **Other questions**
> > >
> > > -- *"I noticed that the probing accuracy for RS-COT and DFS-COT is the same despite a large difference in the information gap."*
> > >
> > > > This is because information gap is not the only quantity that can affect the magnitude of implicit thinking (as measured by a linear probe). For example, the base model’s reasoning pattern can be another factor: as shown in Figure 3, given the same RS-CoT or DFS-CoT supervision, the probe’s accuracy is consistently lower on Qwen2.5-7B (<60%) than on LLaMA3.1-8B (> 80%). Therefore, some models (e.g., Qwen) might inherently prefer explicit thinking and CoTs with relatively large information gap (RS-CoT) is enough to induce explicit, generalizable thinking; while for other models (e.g., LLaMA) might be a more implicit thinker inherently and thus even DFS-CoT with a smaller information gap still cannot suppress its implicit thinking. In summary, such evidence indicates that the magnitude of implicit thinking (as measured by probing accuracy) does not linearly scale with the information gap. The scaling follows a curve similar a pH titration curve, where there exists a critical point. As shown in Table 6, for LLaMA3.1-8B, this critical point comes at a smaller information gap value between 0.097 and 0.05, where its probing accuracy drops from 90% to 53.7% sharply.
> > >
> > > -- *“Do you think of implicit thinking as a binary state? Is there evidence that models can use different degrees of implicit thinking?”*
> > >
> > > > We do believe implicit thinking can be a spectrum metric rather than a binary state. This is actually a very important research question because if a model can implicit think on in-distribution steps seen before to save the costs of search and explicitly search only when necessary, it can achieve OOD generalization with significantly less tokens. To verify this hypothesis, we impose a length penalty on DFS-CoT traces and keep other experimental setups the same as introduced in the paper. The resulting model shows strong implicit thinking on in-distribution (ID) questions (>90% probing accuracy on depth<=5) like the model trained with Forward-CoT. However, this new model also retains the same OOD generalization ability as the original DFS-CoT model trained with no length penalty. Upon closer inspection of their generated search traces, we find that this new model relies mostly on implicit thinking for ID question with depth <= 5 as it barely compute weights of irrelevant boxes (those not on the ground-truth path). For OOD questions with depth k, the model would explicitly search the graph until discovering the (k-5)-th nodes on the ground-truth path and then, utilizing its implicit thinking capable of backtracking the last 5 boxes, computes the remaining 5 boxes with no exploration of irrelevant boxes. This demonstrates a hybrid explicit-implicit reasoning that can more efficiently generalize to OOD questions with deeper graphs. As the Nobel Prize in Physics laureate Chen-Ning Yang said: “learning is a process of turning unknown knowledge into intuition”. Intuition can be accessed implicitly to speed up the reasoning in ID problems but explicit search is necessary to explore OOD, unknown environments so to discover new knowledge. We believe an efficient, generalizable reasoning model should be able to think implicitly and explicitly.
> > >
> > > -- *Typos*:
> > > > We have corrected them in the updated version.

---

### Official Review · Reviewer_vYNE · 2025-11-01

**Soundness:** 2
**Presentation:** 3
**Contribution:** 2
**Rating:** 2
**Confidence:** 4

**Summary:**

This paper takes a closer look at the phenomena of "implicit thinking", where chains-of-thought (CoT) generated by models are post-hoc rationalizations i.e. they they do not causally determine the prediction and the model already has an internal representation of the answer.  To study this, the authors introduce a synthetic graph traversal dataset called WorldOfBoxes (WoB) with two variants:
* WoB-R2L (root-to-leaf): traverse from a root box down to a leaf node
* WoB-L2R (leaf-to-root): start at a leaf node and recover root's value

On top of this dataset they supervise four different chain of thought styles on the same underlying problems in order to determine which styles support out of distribution generalization. The four styles are forward, backtrack, random, and DFS chain of thought. They find that in the R2L setting backtracking based CoT traces generalize best and forward only traces generalize worst. In the L2R setting forward chain of thought appears to do best.  They find that in the R2L setting, backtracking-based CoT traces generalize best and forward-only CoT traces doing worst.

They then measure implicit thinking with a linear probe on frozen model representations. In the R2L setting the probe shows evidence of implicit thinking for models trained with forward chain of thought and some evidence for models trained with DFS CoT. In the L2R setting probe accuracy is high for models trained with backtracking, which they interpret as those models reconstructing the solution structure before emitting the CoT.

The authors then try to formalize some of this in terms of an information gap metric, which aims to measure the completeness of exploration at any point of the CoT trajectory.  If the information gap is large between two consecutive CoT steps, that means there are many possible nodes that can be explored at that step and the current trace has only explored a few by the end.  They show that this information gap quantity correlates with their probe based measure of implicit thinking and use this to motivate their overall hypothesis.

**Strengths:**

Overall, the ideas explored in the paper are interesting. The question of how much additional computation in the form of chain of thought actually contributes to determining the answer is a worthwhile one, and it is good to see it studied in a setting where the underlying structure is fully known.

I liked the synthetic setup. Using a controlled graph traversal task makes the discussion concrete, allows multiple chain of thought styles to be compared on exactly the same problem, and makes it possible to talk precisely about admissible actions, coverage and exploration.

The empirical results in the early part of the paper are interesting. The observation that some supervised traversal styles generalize to deeper trees while others fail, and that this depends on the direction of traversal and the branching factor, is of some value to the community. This part of the work shows that the choice of CoT supervision matters even when the underlying task is fixed.

**Weaknesses:**

My primary criticism of this work is that it is merging several related but distinct questions, and the paper would benefit from a clearer statement of which question it is actually trying to answer. There are at least three threads running through the current version:

1. Faithfulness: Are chains of thought in fact correlated with correct versus incorrect answers, or are they mostly post hoc rationalizations of an already formed answer?

2. Search/exploration: Which traversal style chains of thought actually help with length or depth generalization when the underlying structure is a tree or a graph?

3. Useful metrics for CoT: Can we define a notion of utility or completeness for a chain of thought that tells us when the model was forced to explore versus when it simply followed a narrow path?

Most of the experiments in section 4 are really addressing (2). They show that certain exploration traces, such as backtracking in the root to leaf case, survive depth increases better than forward only traces, and they suggest that this is tied to the branching structure of the graph. I believe some of these results have appeared in the literature (see, eg ref [1]), which makes the search and exploration angle the most solid part of the paper. However, the paper mostly frames itself in the language of (1), using terms such as “implicit thinking” and “post hoc” that belong to the faithfulness discussion. Those two lines of inquiry are related but not identical. If the central question is whether chains of thought are faithful, it is not obvious that length generalization on a synthetic tree task is the most appropriate environment to answer it. Conversely, if the central point is that some distilled search strategies are brittle while others are robust, then the search results should be analyzed more fully.

A second criticism is that the concept of "implicit thinking" is never fully defined.  In the synthetic setting, one could create a working definition such as "the model is implicitly thinking if a linear probe on its hidden state before CoT can classify whether each box is on the GT path."  That is in fact how the paper measures it in practice. If that is the working definition, it is unclear why an additional construct such as the information gap is needed. The current text uses the probe as a litmus test but does not clearly state what hypothesis is being tested, what constitutes strong evidence for implicit thinking, and how this is disentangled from the quality of the supervised search strategy. A weak or mismatched CoT trace will of course force the model to reconstruct missing structure from context, so detecting that the model has done so is not very informative by itself. A more interesting direction, which the current version only hints at, would be to analyze how different traversal algorithms affect convergence or confidence in the target answer.

Also, I am not convinced by the utility of the information gap metric.  It feels redundant with simpler explanations.  In the graph traversal case, one may just as well use a branching ratio or related graph theoretic metrics.  The metric feels ad hoc and hard to transfer for real math reasoning tasks where the admissible action set $C_i$ isn't known.  I think using the probe makes sense as a definition of implicit thinking, but it would have been nice to see some examples in the math reasoning outside of the synthetic setting.  It is also worth noting that everything is dumped into context. So some of what the probe is reading could just be “the model has encoded the structure of this context,” not “the model planned.”  The claim about using DFT to suppress gradients from high gap tokens and hence prevent models from developing implicit thinking is not supported by detailed results. It is not shown whether this improves depth accuracy or only lowers the probe. This part needs numbers.

Though I did not penalize for this, the related work section is not complete. Recent papers that study diversity and coverage in chain of thought should be cited, in particular arXiv 2504.07052, arXiv 2506.05744 and arXiv 2505.21825.

[1] https://arxiv.org/abs/2504.07052v2

[2] https://arxiv.org/abs/2506.05744v3

[3] https://arxiv.org/abs/2505.21825

**Questions:**

1. I am not fully clear on the probing task setup. At what point in the sequence is the probe applied, and what exactly is being classified. Is the probe asked, for every box in the serialized input, to predict whether it is on the ground truth path for the current query? If so, why is the chance baseline 50 percent? In a tree setting the positive class should be the minority unless you balanced it.

---

> ### Author Response · Authors · 2025-11-21
> **Response to Reviewer vYNE**
>
> We sincerely thank the reviewer for their constructive feedback for our work. We especially appreciate that you acknowledge our controllable “synthetic setup” and describe our results regarding “some supervised traversal styles generalize to deeper trees while others fail, and that this depends on the direction of traversal and the branching factor” is “of some value to the community”. We provide our answers to your questions below:
>
> **1. The main storyline of this paper: Search VS Faithfulness**
>
> > The main research question we want to answer in this work is “what factors could affect models’ generalization ability to search from a dependency graph larger than those seen in the training”. And yes, *“Most of the experiments in section 4 are really addressing (2)” (Search/exploration)*. After finding that models trained on CoT with different graph traversal strategies obtain different generalization ability, our next goal is to analyze these models’ behaviors to identify if any associates with their generalization. To this end, we conduct probing experiments and observe a trend: models that generalize worse also tend to implicitly identifies the every neccesary box (as indicated by the stronger linear separability in the final token’s representation before generating CoT). Therefore, the study on implicit thinking and CoT Faithfulness (are they mostly post hoc rationalizations of an already formed answer) is our effort to explain why models trained with some CoT strategies generalize much worse than others by revealing their different reasoning behavior. Finally, to understand this discrepancy of implicit vs explicit reasoning, we propose the Information Gap metric for any CoT traversal of a given dependency graph and show its high correlation with models’ reasoning behavior (implicit vs explicit) and their length generalization.
>
> > We also updated the introduction so that it does not start by discussing the “faithfulness” of CoT but instead opens up with the core research question: length generalization and exploration. We hope this can prevent confusion regarding the main gist of our paper for future readers.
>
> -- *“search results should be analyzed more fully”*
>
> > With the explanation being made above, we also conduct a preliminary analysis on the model-generated search traces and will further expand it in the final version. Specifically, we calculate the percentage $p_2$ of test examples where the model picked the correct 2nd node in CoT (the first node accuracy is universally 100% as it is always the root) when there are the maximum of four children nodes attached to the root node (hence $|C_i|=4$). Intuitively, for a completely explicit-thinking model, $p_2=0.25$ as it has no way to distinguishing between the four candidates at the current step; while for a completely implicit-thinking model who has already pre-planned the CoT steps, $p_2=1$. Our analysis results on models trained with different types of CoT highly align with this intuition. For LLaMA-3.1-8B trained with either Backward-CoT or 16 DFS-CoT traces per sample, $p_2=0.22$; for models trained with either 1 or 3 DFS-CoT traces per sample,  $p_2=0.26$. On the contrary, the model trained with Forward-CoT has $p_2=1$. We further designed some hybrid $x$DFS-$(1-x)$Forward-CoT where at each step, we randomly switch to ForwardCoT with (1-x) probability. For $x=0.95, 0.9, 0.8$, the resulting $p_2$ is $0.32, 0.45, 0.58$ while their OOD accuracy at depth=8 are 49%, 39%, 32%. **The results indicate that the manitude of implicit thinking not only can be quantified by probing, but also can be directly observed in the generated search traces (e.g., by measuring $p_2$), and it negatively correlates with the model’s OOD generalization ability.**
>
> **2. Definition of Implicit Thinking**
>
> > We provide the following definition of implicit thinking on WoB-like problems with a dependency graph of variables (added to the rebuttal revision pdf as Definition 1 (line 354-357):
> >
> > > Definition: Given a dependency graph $g$ with nodes $C$ and the task of finding a path $\tau=[\tau_1, \tau_2, ..., \tau_P]$, we say the model “implicitly thinks” if the representations of $\tau_i\in\tau$: $r(\tau_i|g)$ and $c \in C\setminus\tau$: $r(c|g)$ are linearly separable.
> >
> > Note that the representations $r(\tau_i|g)$ and $r(c|g)$ are only conditioned on the graph, without any generated chain-of-thoughts tokens. Given this definition, it is natural to use linear probe to quantify the magnitude of “implicit thinking” (i.e., the linear separability) within models. The hypothesis we are trying to test is that “the magnitude of implicit thinking is negatively correlated with the model’s ability to generalize to deeper graphs (line 325-327 in the original version and revisited version)”.

---

> > ### Author Response · Authors · 2025-11-22
> > **Response to Reviewer vYNE (cont.)**
> >
> > -- *"A weak or mismatched CoT trace will of course force the model to reconstruct missing structure from context, so detecting that the model has done so is not very informative by itself."*
> >
> > > First, we want to clarify that there is no mismatched CoT trace used in this work. Nor do we consider Forward-CoT as a "weak CoT" with little value of research because, before the Deepseek R1 paper popularizes long and reflective CoTs, most annotated CoTs in reasoning datasets can be seen as Forward-CoT without search/exploration traces. For example, GSM8k’s annotated solutions are Forward-CoTs as only necessary and correct steps are included in the CoT. Our finding suggests that models SFT-ed on such CoTs have limited generalization ability to question with more complex dependencies among variables.
> >
> > > Second, the key takeaway from the probing experiments is not only “suboptimal CoTs force the model to reconstruct missing info from context”, which is briefly discussed in Ye et al., 2024 . More importantly, we show that this implicit thinking ability cannot be generalized to deeper graphs: we can observe from Figure 3 that the probing accuracies of Forward-CoT models fall off the cliff as the depth grows beyond 5. This also explain why models trained with Forward-CoT cannot find the weight of target boxes in dependency graphs deeper than 5 while models trained with other CoT types generalize better.
> >
> > > Ye, Tian, et al. "Physics of language models: Part 2.1, grade-school math and the hidden reasoning process." arXiv preprint arXiv:2407.20311 (2024).
> >
> > **The Information Gap Hypothesis**
> >
> > -- *"In the graph traversal case, one may just as well use a branching ratio or related graph theoretic metrics."*
> >
> > > Information Gap is a metric for graph traversal strategy (e.g., forward-only, random search, depth-first search) while branching ratio is a property for the graph itself. Given the same graph with a fixed branching ratio, there exists different graph traversal strategies bearing different information gap, and our experiments show that they lead to different length generalization ability (Figure 4).
> > >
> > > According to the definition, the per-step Information Gap is $\mathrm{log} \frac{|C_i \cap \tau|}{|C_i|}$. Therefore, it measures not only the traversal’s branching ratio (controlled by the denominator $|C_i|$), but also its **completeness**: at step i, the numerator $|C_i\cap \tau|$ considers the entire CoT trajectory including nodes explored in the future steps $>i$. Hence, given a fixed branching ratio ($|C_i|$), $q^{\phi}(\tau_i|\tau)$ might have different value depending on how many nodes in $C_i$ are explored by the end of $\tau$ (i.e., the completeness).
> >
> > -- *"The metric feels ad hoc and hard to transfer for real math reasoning tasks where the admissible action set isn't known."*
> > > We acknowledge this as a limitation of the information gap metric, that it cannot be applied for real-world tasks without a known dependency graph of variables. However, as Reviewer yrFH acknowledged, this limitation is “inherent to its (very strong) controlled methodology”. Furthermore, we provided a blueprint for measuring the “Information Gap within a CoT in the wild”  (line 454-458 in the original version and line 475-459 in the revisitedd version). Intuitively, this definition suggests that a large information gap is caused by low-likelihood, off-policy tokens in CoT. We updated the paper to show improved generalization and less implicit thinking after applying the DFT loss (Table 7). This generalized Information Gap metric also resonates with the findings of a number of recent work other than the DFT paper, which show improved math reasoning by down-scaling or down-sampling the off-policy CoT (i.e., high-information-gap CoT) in a hybrid SFT-RL setup (Fu et al., 2025, Yan et al., 2025).
> >
> > > Fu, Yuqian, et al. "SRFT: A Single-Stage Method with Supervised and Reinforcement Fine-Tuning for Reasoning." arXiv preprint arXiv:2506.19767 (2025).
> > >
> > > Yan, Jianhao, et al. "Learning to reason under off-policy guidance." arXiv preprint arXiv:2504.14945 (2025).
> >
> >
> > **Missing Related Work**
> >
> > > Thank you for the suggestions! We have added them to the paper.

---

### Author Response · Authors · 2025-12-04
**Summary for Area Chair: Rebuttal Overview**

### Reviewer Consensus on Strengths

Reviewers converge on several key strengths of our work:

> 1. **Novel and Important Problem**: Both fiU3 and yrFH acknowledge our investigation of implicit thinking and length generalization in CoT reasoning as timely and well-motivated
>
> 2. **Strong Experimental Design**: yrFH rates our experimental design as "excellent," noting that the WOB synthetic dataset allows us to "control for all confounding variables" and provides a "clever way to isolate different types of reasoning traces" (fiU3)
>
> 3. **Novel Contribution**: Reviewers recognize the "Information Gap" concept as novel and valuable - yrFH describes it as providing "a compelling explanation for why models take implicit shortcuts," while fiU3 acknowledges we show "significant evidence that implicit thinking is associated with worse length generalization"
>
> 4. **Clear Presentation**: yrFH rates presentation as "excellent" with the paper being "exceptionally clear" and "easy to follow"

### Key Concerns and Our Responses

> 1. Formalization of Implicit Thinking (fiU3)
> - **Concern**: Lacked explicit definition; measurement via linear probing needed formalization
>- **Resolution**: Added formal mathematical definition (Definition 1 in line 355-357 in the revision draft) showing implicit thinking occurs when representations $r(\tau_i|g)$ and $r(c|g)$ are linearly separable, clarifying it's based solely on graph structure without generated tokens
>
> 2. Information Gap Experimental Evidence (fiU3)
> - **Concern**: Only 5 data points in original Figure 4
> - **Resolution**: Extended to 8 data points by adding 3 hybrid CoT types (xDFS+(1-x)Fwd-CoT where x=0.95, 0.9, 0.8), confirming the negative correlation between Information Gap and OOD generalization
>
> 3. Mathematical Intuition for Information Gap Mechanism (fiU3)
> - **Concern**: Why Information Gap is the relevant variable
> - **Resolution**: Provided detailed mathematical explanation showing Information Gap represents the "residual ambiguity" - the explicit-exploration penalty not explained by trajectory-level supervision, which must be resolved by implicit thinking
>
> 4. Generalization Beyond WOB (yrFH & fiU3)
> - **Concern**: Core claims validated only on synthetic dataset
> - **Resolution**:
>>  - Cited Liu et al. (2025) NVIDIA study showing SFT dataset scaling principles transfer to real benchmarks (AIME 25, LiveCodeBench)
>>  - Updated references to DFT loss work (Fu et al., Yan et al. 2025) showing improved math reasoning through down-sampling high-information-gap CoT
>
> 5. Model Scale Effects (yrFH)
> - **Concern**: Do findings hold for larger models (70B+)?
> - **Resolution**: Added results from Qwen2.5-3B and Qwen2.5-14B showing positive scaling in OOD length generalization and negative scaling in implicit thinking (measured by probing accuracy)

---

We have incorporated all clarifications and additional experiments into the revised manuscript, addressing all reviewers' concerns while maintaining the paper's core contributions.

---

### Meta-Review · Area_Chair_EaFH · 2026-01-06

**Summary:**

The authors propose "WorldOfBoxes" (WoB), a synthetic graph traversal benchmark, to investigate the phenomenon of "implicit thinking" in Chain-of-Thought (CoT) reasoning. They hypothesize that implicit thinking---where models internally pre-compute solutions before generating reasoning traces---is driven by a high "Information Gap" in the training supervision. While the paper provides a clean experimental setup to isolate these variables, the reliance on synthetic data and the limited transferability of the proposed metric raise significant concerns regarding the work's practical significance and applicability to real-world Large Language Model (LLM) reasoning tasks.

The submission presents a scientifically sound study within a highly controlled synthetic environment. However, the Area Chair finds that the gap between this synthetic setting and real-world application is too large. The core contribution---the Information Gap metric---cannot be easily calculated or applied to standard reasoning benchmarks, limiting its utility for the community. Furthermore, the skepticism from Reviewer vYNE regarding the narrative coherence and the novelty of the search findings (vs. existing literature) is well-founded. Therefore, the paper does not currently meet the bar for acceptance.

**Reviewer Concerns:**

**Addressed by Rebuttal**:
- Formalization (fiU3): The authors added a formal definition of implicit thinking based on the linear separability of node representations.
- Experimental Robustness (fiU3): Additional data points were provided for the Information Gap correlation in Figure 4.
- Scaling (yrFH): Experiments with Qwen2.5-3B and 14B were added to show scaling trends.

**Outstanding Concerns**:
 -  Real-World Applicability and Metric Utility (vYNE, yrFH): The primary critique remains the disconnect between the synthetic WoB environment and realistic reasoning tasks. Reviewer vYNE correctly identifies that the "Information Gap" metric is theoretically interesting but practically limited because it requires a known dependency graph (available in WoB but unknown in tasks like MATH or GSM8K). The authors' rebuttal suggestion to use ``low-likelihood tokens'' as a proxy is speculative and lacks the rigorous validation required to prove utility beyond the synthetic sandbox.
  - Scope and Narrative (vYNE): The paper attempts to bridge distinct concepts---faithfulness, search/exploration, and supervision metrics---but results in a disjointed narrative. The findings on ``implicit thinking'' are tightly coupled to the specific artifacts of the WoB dataset. Without demonstrating that this mechanism is a dominant factor in general purpose LLMs on natural data, the claims feel over-generalized.
   - Significance: While Reviewer yrFH praised the controlled methodology, the consensus is that the ``clean'' nature of the experiment does not compensate for the lack of external validity. For a top-tier venue, a paper must demonstrate that its insights transfer to the broader domain, which is not sufficiently established here.

**Reviewer Scores:**

- Reviewer vYNE (2): Likely remains at 2. The fundamental objection regarding the ``ad hoc'' nature of the metric and the merging of distinct research questions was not adequately resolved by the rebuttal.
- Reviewer fiU3 (4): Likely remains at 4. While the formalization was improved, the reviewer's concern about the limited evidence for the Information Gap mechanism and its broader relevance persists.
- Reviewer yrFH (8): Likely drops to 6. While the reviewer appreciated the experimental rigor, the validity of the concerns raised by other reviewers regarding the exclusive reliance on synthetic data highlights a significant limitation in the paper's contribution.

---

### Decision · Program_Chairs · 2026-01-26

Reject